# Impact of calculated plasma volume status on all-cause and cardiovascular mortality: 4-year nationwide community-based prospective cohort study

Yoichiro Otaki[1], Tetsu Watanabe[1]*, Tsuneo Konta[1], Masafumi Watanabe[1], Koichi Asahi[2], Kunihiro Yamagata[2], Shouichi Fujimoto[2], Kazuhiko Tsuruya[2], Ichiei Narita[2], Masato Kasahara[2], Yugo Shibagaki[2], Kunitoshi Iseki[2], Toshiki Moriyama[2], Masahide Kondo[2], Tsuyoshi Watanabe[2]

1 Department of Cardiology, Pulmonology, and Nephrology, Yamagata University School of Medicine, Yamagata, Japan, 2 Steering Committee of Research on Design of the Comprehensive Health Care System for Chronic Kidney Disease (CKD) Based on the Individual Risk Assessment by Specific Health Check, Fukushima, Japan

* tewatana@med.id.yamagata-u.ac.jp

**Data Availability Statement:** Data cannot be shared publicly due to ethical restrictions on sharing data publicly. The protocol of this project

## Abstract

### Background

Plasma volume status (PVS), a marker of plasma volume expansion and contraction, is gaining attention in the field of cardiovascular disease because of its role in the prevention and of the management of heart failure. However, it remains undetermined whether an abnormal PVS is a risk for all-cause and cardiovascular mortality in the general population.

### Methods and results

We used a nationwide database of 230,882 subjects (age 40–75 years) who participated in the annual "Specific Health Check and Guidance in Japan" check-up between 2008 and 2011. There were 586 cardiovascular deaths, 2,552 non-cardiovascular deaths, and 3,138 all-cause deaths during the follow-up period of four years. Abnormally high and low PVS were identified from the results of 80% of all subjects (high and low PVS ≥ 7 and < -13.3, respectively). Multivariate Cox proportional hazard regression analysis demonstrated that high PVS was an independent risk factor for all-cause, cardiovascular and non-cardiovascular deaths. Although low PVS was a positive risk factor for cardiovascular deaths as well, it was a negative risk factor for non-cardiovascular deaths. The addition of PVS to cardiovascular risk factors significantly improved the C-statistic, net reclassification, and integrated discrimination indexes.

### Conclusions

This is the first prospective report to reveal the impact of PVS on all-cause and cardiovascular mortality. PVS could be an additional risk factor for all-cause and cardiovascular mortality in the general population.

(Research on the Positioning of Chronic Kidney Disease in Specific Health Check and Guidance in Japan) determined that analytical data were distributed only to the members of steering committee because the data contain potentially identifying information. Data are available upon requestfrom the Department of Chronic Kidney Disease Initiatives; Fukushima Medical University School of Medicine; 1-Hikarigaoka, Fukushima 960-1295, Japan; Phone & Fax: +81-24-547-1898; E-mail dckdi@fmu.ac.jp.

**Funding:** The authors received no specific funding for this work.

**Competing interests:** The authors have declared that no competing interests exist.

## Introduction

Regulation of plasma volume is important in pregnancy and is a therapeutic target in sepsis and heart failure [1–3]. Heart failure remains a major and increasing public health problem, with a high mortality rate [4]. It was reported that imbalanced volume homeostasis causes systemic congestion and peripheral and pulmonary edema in heart failure [5]. Plasma volume expansion underlies systemic congestion, which is a well-known, clinically, and prognostically relevant complication of heart failure [6]. Since accurate measurement of plasma volume is technically difficult and invasive as its determination requires pulmonary artery catheterization or administration of tracer molecules [7–9], several formulae have been derived from routinely collected clinical data to calculate estimates of plasma volume. It was reported that plasma volume calculated using these formulae is a useful predictor of clinical outcome in patients with heart failure [10,11]. However, these estimates of plasma volume were reportedly poorly correlated with measured plasma volume [12].

Recently, plasma volume status (PVS), an index of the degree to which patients have deviated from their ideal plasma volume, is gaining attention in patients with heart failure. PVS is associated with cardiac events and mortality in patients with heart failure [13–15]. American College of Cardiology/American Herat Association guidelines have recommended volume status be assessed [16]. However, the impact of PVS on all-cause and cardiovascular death in the general population remains unknown. Thus, we hypothesized that PVS may serve as an early identification of high-risk subjects for all-cause and cardiovascular deaths in the general population. The present study aimed to examine whether PVS is a novel risk factor for all-cause and cardiovascular deaths in the general population.

## Method

### Ethics statement

All procedures performed in studies involving human participants were undertaken in accordance with the ethical, institutional, and/or national research committee guidelines of the centers at which the studies were conducted (Yamagata University, 2008, no. 103) and in compliance with the 1964 Helsinki declaration and its later amendments or comparable ethical standards. The institutional ethics committee of Yamagata University School of Medicine approved the study.

This study was performed according to the Ethical Guidelines for Medical and Health Research Involving Human Subjects enacted by the Ministry of Health, Labour and Welfare of Japan (http://www.mhlw.go.jp/file/06-Seisakujouhou-10600000-Daijinkanboukouseikagakuka/0000069410.pdf; http://www.mhlw.go.jp/file/06-Seisakujouhou-10600000-Daijinkanboukouseikagakuka/0000080278.pdf). In the context of the guideline, the investigators shall not necessarily be required to obtain informed consent, but we publicized information concerning this study on the web (http://www.fmu.ac.jp/univ/sangaku/data/koukai_2/2771.pdf) and ensured that there was an opportunity for the research subjects to decline the use of their personal information. All data were fully anonymized.

### Study population

This study is a part of an ongoing "Research on design of the comprehensive health care system for chronic kidney disease (CKD)" based on individual risk assessments by the Specific Health Check-up for all inhabitants of Japan between the ages of 40 and 74 years and is covered by the Japanese national health insurance. We utilized data obtained from the following 16 prefectures (i.e., administrative regions): Hokkaido, Tochigi, Saitama, Chiba, Nagano,

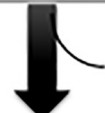

Research on design of the comprehensive health care system for chronic kidney disease (CKD)

Calculated plasma volume status
n=230,989

107 due to the lack of essential data

n=230,882 (88,775 men and 142,107 women)

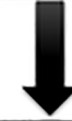

914,292 person-years
(Median follow up period: 4 years)

<Endpoints>
All-cause deaths: 3,138
Cardiovascular deaths: 586
Non-cardiovascular deaths: 2,552

**Fig 1. A flow chart of the study selection process.**

Niigata, Ishikawa, Fukui, Gifu, Hyogo, Tokushima, Fukuoka, Saga, Nagasaki, Kumamoto, and Okinawa. These prefectures were divided into four region areas; Hokkaido and Tohoku; Kanto and Koshinetsu; Kinki, Shikoku, and Chugoku; and Kyushu and Okinawa. A flow chart of the selection process used in the study is shown in Fig 1. We collected data from 230,989 subjects (aged 40–74 years) who participated in the health check-ups of 2008–2011. Among them, 107 were excluded from this study due to lack of essential data. Therefore, 88,775 men and 142,107 women were included in this study.

## Definition of cardiovascular risks

Blood pressure was measured the following method [17]. Participants were seated with back supports. After resting for at least 5 minutes, blood pressure was measured 2 times without conversation. Blood pressure was determined by an average of 2 blood pressure readings. Hypertension was defined as a systolic blood pressure $\geq$ 140 mmHg, diastolic blood pressure $\geq$ 90 mmHg, or antihypertensive medication use. Diabetes mellitus was defined as a fasting blood sugar (FBS) $\geq$ 126 mg/dL, glycosylated haemoglobin A1c (HbA1c) $\geq$ 6.5% (National Glycohemoglobin Standardization Program), or anti-diabetic medication use. Dyslipidemia was defined as high-density lipoprotein cholesterol (HDL-C) < 40 mg/dL, low-density lipoprotein cholesterol (LDL-C) $\geq$ 140 mg/dL, triglyceride $\geq$ 150 mg/dL, or lipid-lowering medication use.

## Definition of PVS

Actual PV (aPV), ideal PV (iPV), and PVS were calculated by the following equations: aPV = (1-hematocrit)×[a+(b×weight (kg))] where hematocrit is a fraction (Men: a = 1530 and b = 41; women, a = 864 and b = 47.9); iPV = c×weight (kg) where c = 39 in men and c = 40 in women; and PVS = [(aPV-iPV)/iPV]×100% [14,15,18]. Normal range of PVS has not been defined yet. Therefore, we defined abnormally high and low PVS as PVS $\geq$ 7 and < -13.3, respectively, based on the results for 80% of all subjects. Since aPV is often under iPV, the value of PVS

could become less than 0. In the Treatment of Preserved Cardiac Function Heart Failure with an Aldosterone Antagonist Trial (TOPCAT) study, PVS was reported to be less than 0 in 91% of heart failure patients [13]. High and low PVS are considered as plasma volume expansion and contraction, respectively.

## Measurements

FBS, HbA1c, total cholesterol, HDL-C, LDL-C and triglyceride levels were measured. All blood and urine analyses were performed at a local laboratory. The methods for the analyses were not standardized between laboratories. However, the analyses were based on the Japan Society of Clinical Chemistry recommended methods for laboratory tests, which have been widely accepted by laboratories across Japan.

## Endpoint and follow-up

After obtaining permission from the Ministry of Health, Labour and Welfare, we accessed the database containing death certificates for all deaths that occurred between 2008 and 2015. All subjects were prospectively followed for a median follow up period of 4 years (interquartile range, 2.9–5.2 years; longest follow up, 7 years). The endpoints were cardiovascular death, non-cardiovascular death, and all-cause death. The cause of death was determined by reviewing the death certificates and classified based on the death code (International Classification of Diseases, 10th Revision).

## Statistical analysis

Normality of continuous variables was checked by a Kolmogorov-Smirnov-Lillefors test. Subjects without essential data were excluded from this study. Continuous and categorical variables were compared with t-tests and chi-square tests, respectively. Survival curves were constructed using the Kaplan-Meier method and compared using log-rank tests. A Cox proportional hazard analysis was performed to determine independent predictors for all-cause death, and significant predictors selected in univariate analysis were entered into the multivariate analysis. Receiver operating characteristics (ROC) curves for all-cause deaths, cardiovascular and non-cardiovascular deaths were constructed and used as a measure of the predictive accuracy of PVS for all-cause deaths. We calculated the net reclassification index (NRI) and integrated discrimination index (IDI) to measure the quality of improvement for the correct reclassification by the addition of PVS to the multivariate model. Values of $P < 0.05$ were considered statistically significant. All statistical analyses were performed using standard statistical program packages (JMP version 12, SAS Institute Inc., Cary, NC, USA; and R 3.0.2 with additional packages including Rcmdr, Epi, pROC, and PredictABEL).

## Results

### Baseline characteristics and comparison of clinical characteristics between subjects with high and low PVS

The subjects' baseline characteristics are shown in Table 1.

Hypertension, dyslipidemia and diabetes mellitus were identified in 104,003 (45%), 127,474 (55%), and 20,103 (8.7%) subjects, respectively. The mean PVS was -2.7%. The subjects were divided into three groups based on the PVS: low PVS group, PVS < -13.3, n = 22,613; normal PVS group, PVS from -13.3 to 6.9, n = 185,415; and high PVS group, PVS ≧7, n = 22,854.

Subjects with high PVS were older and had lower prevalence of dyslipidemia than those in the other two groups. Subjects with high PVS showed lower levels of body mass index, systolic

**Table 1. Comparison of clinical characteristics between subjects with low, normal and high PVS.**

| Variables | All subjects n = 230,882 | Low PVS n = 22,613 | Normal PVS n = 185,415 | High PVS n = 22,854 |
|---|---|---|---|---|
| Age, years | 64 ± 8 | 62 ± 9* | 64 ± 8 | 65 ± 9*† |
| Male, n (%) | 88,775 (38%) | 16,203 (72%) | 63,772 (34%) | 8,800 (39%)‡ |
| BMI, kg/m$^2$ | 22.9 ± 2.9 | 26.6 ± 3.5* | 22.8 ± 2.9 | 20.0 ± 2.7*† |
| Hypertension, n (%) | 104,003 (45%) | 13,496 (59%) | 82,371 (44%) | 8,136 (36%)‡ |
| Systolic BP, mmHg | 129 ± 17 | 134 ± 17* | 128 ± 17 | 124 ± 18*† |
| Diastolic BP, mmHg | 76 ± 11 | 82 ± 11* | 76 ± 11 | 72 ± 11*† |
| Dyslipidemia, n (%) | 127,474 (55%) | 16,104 (71%) | 103,233 (56%) | 8,137 (36%)‡ |
| Diabetes mellitus, n (%) | 20,103 (8.7%) | 3,345 (14.8%) | 14,811 (8.0%) | 1,947 (8.5%)‡ |
| Smoking, n (%) | 31,694 (14%) | 5,990 (27%) | 22,773 (12%) | 2,931 (13%)‡ |
| Region area | | | | |
| Hokkaido and Tohoku | 22,314 (10) | 3,034 (13%) | 17,959 (10%) | 1,321 (6%)‡ |
| Kanto and Koshinetsu | 104,487 (45) | 9,967 (44%) | 84,961 (45%) | 9,829 (43%) |
| Kinki, Shikoku and Chugoku | 15,038 (7) | 1,635 (7%) | 12,417 (7%) | 986 (4%) |
| Kyushu and Okinawa | 89,043 (38) | 7,977 (35%) | 70,348 (38%) | 1,0718 (47%) |
| *Biochemical data* | | | | |
| PVS | -2.7 ± 7.5 | -17.1 ± 3.8* | -3.5 ± 5.1 | 18.7 ± 18.5*† |
| RBC, 10$^4$/μL | 431 ± 68 | 495 ± 56* | 435 ± 53 | 335 ± 141*† |
| Hb, g/dL | 13.4 ± 1.6 | 15.5 ± 1.0* | 13.5 ± 1.1 | 10.2 ± 4.1*† |
| Hematocrit, mg/dL | 40.6 ± 4.8 | 47.2 ± 2.9* | 41.0 ± 3.0 | 31.2 ± 12.2*† |
| eGFR, ml/min/1.73m$^2$ | 75.7 ± 17.0 | 74.5 ± 15.6* | 75.7 ± 16.7 | 76.6 ± 19.6*† |
| HbA1c (%) | 5.4 ± 0.7 | 5.6 ± 0.9* | 5.4 ± 0.7 | 5.3 ± 0.7*† |
| FBS, mg/dL | 97 ± 21 | 105 ± 28* | 97 ± 20 | 95 ± 21*† |
| Total cholesterol, mg/dL | 211 ± 36 | 218 ± 38* | 212 ± 36 | 196 ± 35*† |
| Triglyceride, mg/dL | 120 ± 79 | 163 ± 107* | 119 ± 76 | 94 ± 62*† |
| HDL-C, mg/dL | 62 ± 17 | 55 ± 14 | 63 ± 17 | 65 ± 17 |
| LDL-C, mg/dL | 125 ± 31 | 131 ± 32* | 126 ± 31 | 112 ± 30*† |
| *Medications* | | | | |
| Anti-hypertensive drug, n (%) | 68,615 (30%) | 8,435 (37%) | 54,686 (29%) | 5,494 (24%)‡ |
| Anti-diabetic drug, n (%) | 11,853 (5.1%) | 1,539 (6.8%) | 9,006 (4.9%) | 1,308 (5.7%)‡ |
| Anti-dyslipidemia drug, n (%) | 37,629 (16.3%) | 3,723 (16.5%) | 31,367 (16.9%) | 2,539 (11.1%)‡ |

Data are expressed as mean ± SD, number (percentage), or median.

BMI, body mass index; BP, blood pressure; eGFR, estimated glomerular filtration rate; FBS, fasting blood sugar; HbA1c, glycosylated hemoglobin A1c; HDL-C, high-density lipoprotein cholesterol; LDL-C, low-density lipoprotein cholesterol; PVS, plasma volume status; RBC, red blood cell count.

* P<0.05 vs. normal PSV group,

† P<0.05. vs. low PSV group by analysis of variance (ANOVA) with Tukey's post hoc test,

‡ P<0.05 by chi-square test.

and diastolic blood pressure, red blood cell count, haemoglobin, FBS, HbA1c, total cholesterol, and triglyceride and LDL-C and higher level of eGFR than those without it (Table 1).

Subjects with low PVS were younger and more likely to be male; to have hypertension, dyslipidemia, or diabetes mellitus; to be current smokers; or be taking anti-hypertensive or anti-diabetic drugs than those with normal or high PVS. Subjects with low PVS showed higher body mass index, systolic and diastolic blood pressure, red blood cell count, haemoglobin, FBS, HbA1c, total cholesterol, and triglyceride and LDL-C levels and lower eGFR (Table 1).

It was reported that high altitude affects volume status and its regulator hormone secretion [19], indicating the region difference in PVS. There was a significant difference in the prevalence of high and low PVS among region areas.

## PVS and mortality

All subjects were prospectively followed during a median follow-up period of 4 years. During the follow-up period, there were 586 cardiovascular deaths, 2,552 non-cardiovascular deaths, and 3,138 all-cause deaths.

Since it was reported that there was a non-linear relationship between PVS and mortality in patients with heart failure [14,18], we examined the unadjusted hazard ratio for subject groups stratified by 1% increments of PVS. As shown in Fig 2, a consistently significant higher risk was seen in the groups with PVS > 5% or < -20% compared with the group with PVS between -6% and -5.1% who had the lowest risk (referent group).

Kaplan-Meier analysis demonstrated that subjects with high PVS had higher rates of all-cause, cardiovascular, and non-cardiovascular deaths than those without it (Fig 3A, 3B and 3C). On the other hand, subjects with low PVS had a higher rate of cardiovascular deaths compared to normal PVS group, whereas there were no significant differences in the non-cardiovascular and all-cause mortalities between subjects with low and normal PVS.

To determine the risk factors for predicting all-cause, cardiovascular, and non-cardiovascular deaths, we performed univariate and multivariate Cox proportional hazard regression analyses. In the univariate analysis, high PVS was significantly associated with all-cause, cardiovascular and non-cardiovascular deaths (Table 2).

In addition, low PVS was significantly associated with all-cause and cardiovascular deaths, but not non-cardiovascular deaths. The multivariate Cox proportional hazard regression

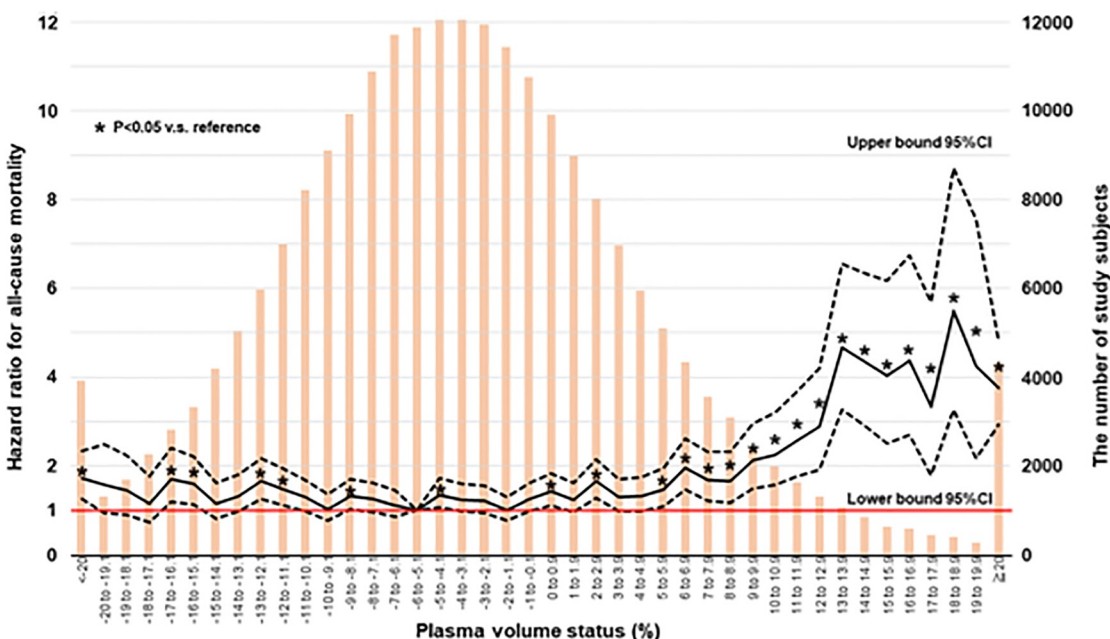

**Fig 2. The unadjusted hazard ratio for subject groups stratified by 1% increments of PVS.** CI, confidence interval; PVS, plasma volume status. Solid line shows the hazard ratios for subject groups stratified by 1% increments of PVS. Dotted lines show the 95% confidence interval for subjects stratified by 1% increments of PVS. Orange bar shows the distribution of study subjects. The referent group was defined as the lowest risk group. *P<0.05 v.s. referent group.

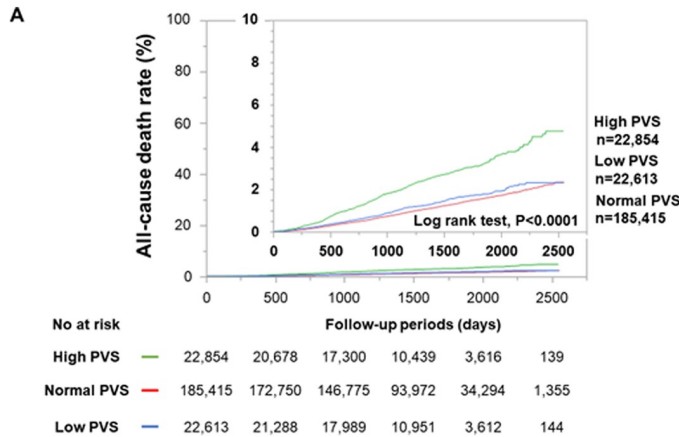

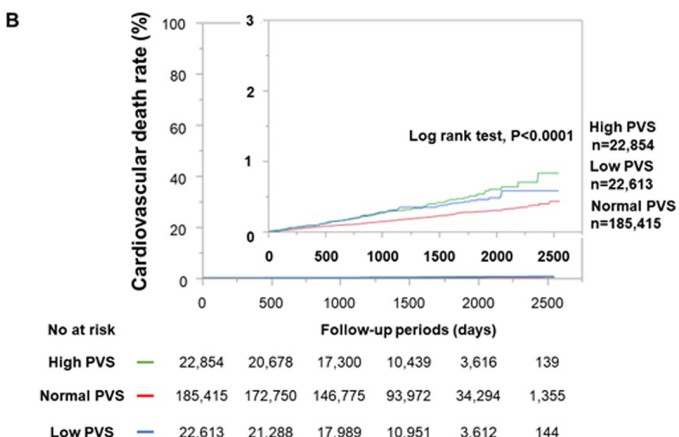

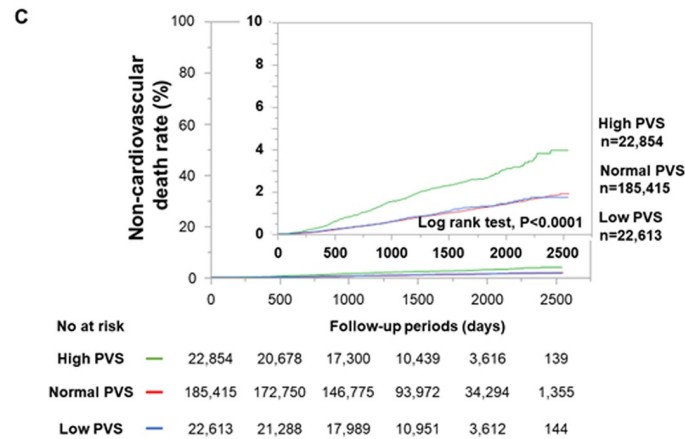

**Fig 3. Kaplan-Meier analysis of all-cause deaths (A), cardiovascular deaths (B), and non-cardiovascular deaths (C) among subjects with low, normal, and high PVS.** PVS, plasma volume status. Green, red, and blue lines show the survival curves for high, normal, and low PVS groups, respectively.

**Table 2. Univariate and multivariate Cox proportional hazard analyses of predicting all-cause, cardiovascular, and non-cardiovascular deaths.**

| Variables | Univariate analysis | | | Multivariate analysis* | | |
|---|---|---|---|---|---|---|
| | HR | 95%CI | P value | HR | 95%CI | P value |
| *All-cause deaths* | | | | | | |
| High vs. normal PVS | 2.186 | 1.993–2.394 | <0.0001 | 2.041 | 1.856–2.239 | <0.0001 |
| Low vs. normal PVS | 1.161 | 1.030–1.305 | 0.0150 | 0.938 | 0.829–1.057 | 0.2991 |
| *Cardiovascular deaths* | | | | | | |
| High vs. normal PVS | 1.852 | 1.466–2.315 | <0.0001 | 1.823 | 1.437–2.290 | <0.0001 |
| Low vs. normal PVS | 1.674 | 1.314–2.107 | <0.0001 | 1.332 | 1.037–1.691 | 0.0251 |
| *Non-cardiovascular deaths* | | | | | | |
| High vs. normal PVS | 2.262 | 2.044–2.498 | <0.0001 | 2.085 | 1.880–2.307 | <0.0001 |
| Low vs. normal PVS | 1.045 | 0.909–1.196 | 0.5253 | 0.848 | 0.735–0.973 | 0.0191 |

CI, confidence interval; HR, hazard ratio; PVS, plasma volume status.

*after adjustment for age, sex, hypertension, diabetes mellitus, dyslipidemia, smoking, and region area.

analysis demonstrated that high PVS was an independent predictor of future all-cause, cardiovascular and non-cardiovascular deaths after adjustment for age, sex, hypertension, dyslipidemia, diabetes mellitus, smoking, and region area (Table 2). On the other hand, low PVS was a positive risk factor for cardiovascular deaths, but a negative risk factor for non-cardiovascular deaths after adjustment for age, sex, hypertension, dyslipidemia, diabetes mellitus, smoking, and region area (Table 2).

## Improvement of reclassification by addition of PVS to predict all-cause, cardiovascular, and non-cardiovascular mortality

To examine whether model fit and discrimination improve with addition of PVS to the basic predictors such as age, sex, hypertension, dyslipidemia, diabetes mellitus, smoking, and region area, we evaluated the improvement of C index, NRI and IDI. The ROC curve analyses demonstrated that the C indices of the baseline model for all-cause mortality, cardiovascular mortality, and non-cardiovascular mortality were significantly improved by the addition of PVS. NRI and IDI were also significantly improved by the addition of PVS (Table 3).

**Table 3. Statistics for model fit and improvement with the addition of PVS on the prediction of all-cause, cardiovascular, and non-cardiovascular death.**

| | C index | NRI (95%CI, P value) | IDI (95%CI, P value) |
|---|---|---|---|
| *All-cause mortality* | | | |
| Baseline model | 0.6823 | Reference | Reference |
| Baseline model+PVS | 0.6928 | 0.0464 | 0.0015 |
| | (P<0.0001) | (0.0318–0.0610, P<0.0001) | (0.0012–0.0018, P<0.0001) |
| *Cardiovascular mortality* | | | |
| Baseline model | 0.7177 | Reference | Reference |
| Baseline model+PVS | 0.7244 | 0.0470 | 0.0003 |
| | (P = 0.0264) | (0.0155–0.0786, P = 0.0035) | (0.0001–0.0004, P<0.0001) |
| *Non-cardiovascular mortality* | | | |
| Baseline model | 0.6762 | Reference | Reference |
| Baseline model+PVS | 0.6882 | 0.0567 | 0.0014 |
| | (P<0.0001) | (0.0398–0.0736, P<0.0001) | (0.0011–0.0017, P<0.0001) |

Baseline model includes age, gender, hypertension, diabetes mellitus, dyslipidemia, smoking, and region area.

IDI, integrated discrimination index; NRI, net reclassification index; 95%CI, 95% confidence interval.

## Discussion

The main findings in the present study were as follows: (1) A J-curve association of PVS with all-cause mortality; (2) Kaplan-Meier analysis demonstrated that subjects with high PVS had higher rates of all-cause, cardiovascular, and non-cardiovascular deaths and subjects with low PVS had higher rate of cardiovascular deaths compared to those with normal PVS; (3) multivariate analysis demonstrated that high PVS was an independent predictor of all-cause, cardiovascular, and non-cardiovascular deaths and low PVS was an independent predictor of cardiovascular deaths; (4) the addition of PVS to other risk factors improved the prediction of all-cause, cardiovascular, and non-cardiovascular deaths in the general population.

The prognostic value of PVS has never been examined in the general population until now. The present study extended the past studies regarding calculated plasma volume and can bring new insight into the possibility that PVS could be a feasible marker for early identification of high-risk subjects in the general population. The clinical application of PVS is mainly discussed in the field of heart failure.

The data obtained from Valsartan in Heart Failure Trial (Val-HeFT) indicated that PVS was associated with increased mortality and first morbid events in J-curve fashion with the highest risk seen with PVS > -4 in patients with symptomatic heart failure [14]. Peter et al reported that PVS of -6.5% optimally predicted absence of adverse outcomes, and the rate of cardiac events were increased with advancing plasma expansion in patients with heart failure with reduced ejection fraction and those with mid-range ejection fraction [18]. The TOPCAT study demonstrated that increment in PVS is associated with a higher risk of all-cause death and heart failure hospitalization in patients with heart failure with preserved ejection fraction [13]. Taking these results into consideration, PVS could be a useful predictor of poor clinical outcome in patients with heart failure independently of ejection fraction. Annette et al reported that PVS greater than -5.6% is associated with adverse inpatient outcomes such as in-hospital death, postoperative complications and prolonged hospitalization in patients undergoing coronary bypass graft surgery [15]. Interestingly, their cut-off value for the absence of adverse outcomes was -5.6%. In accordance with these reports, our results from groups stratified by 1% increments of PVS showed that PVS of -6 to -5.1 best predicted the absence of all-cause deaths and J-curve association of PVS with all-cause mortality was observed in the general population. These findings indicated that imbalanced PVS may contribute to the development of cardiovascular disease.

The precise mechanism by which low PVS was associated with cardiovascular mortality is unclear. There is a close relationship between plasma volume and the renin angiotensin aldosterone system. It is well known that renin secretion from juxta glomerular cells is augmented by reduction in plasma volume, leading to renin angiotensin aldosterone system activation, in an attempt to reduce renal excretion of sodium, thus tending to restore plasma volume by increasing blood osmolality [20]. Subjects with low PVS had a higher prevalence rate of hypertension and higher levels of systolic and diastolic blood pressure than those with normal and high PVS. These results suggested the possibility that volume contraction may worsen cardiovascular mortality through activation of the renin angiotensin aldosterone system. In addition, subjects with low PVS accompanied higher prevalence of smoking, suggesting the presence of erythrocytosis secondary to smoking, which poses the risk of thrombosis. Therefore, thromboembolic events may contribute to the high cardiovascular mortality in subjects with low PVS.

Also, the possible pathophysiological mechanism by which high PVS may contribute to the increase in the risk of all-cause, cardiovascular, and non-cardiovascular mortality is unclear. It was reported that low hematocrit and anemia were associated with death in patients with lung cancer, which is the principle cause of cancer death in Japan [21,22]. These reports supported

our result that high PVS was associated with non-cardiovascular deaths in the general population. With regard to cardiovascular mortality, previous reports discussed that neurohumoral activation such as that of the renin angiotensin aldosterone system and sympathetic nervous activation exacerbates cardiac function, leading to poor prognosis in heart failure patients [23]. In healthy subjects, as opposed to heart failure patients, high plasma volume potentially inhibits renin secretion. There must be a different mechanism operating in the general population. Potential explanation is that volume overload caused by plasma volume expansion may exacerbate cardiac function [6]. Since this study is a prospective observational study, it is beyond the scope of the study to determine the association between high PVS and mortality in the general population.

The clinical counterpart of the present study was that high PVS is associated with all-cause death, indicating the fact that subjects with high PVS need further examination for fatal disease such as cancer, cardiovascular disease, and pulmonary disease. On the other hand, subjects with low PVS need examination to exclude cardiovascular disease. Importantly, addition of PVS to the established risk factors improved c-statistics, NRI and IDI, indicating that it could be useful clinical information for the prevention, early identification and management of potentially fatal disease. Interestingly, it has been reported that renin angiotensin aldosterone inhibitors optimize plasma volume in patients with heart failure [18]. In addition, sodium-glucose cotransporter 2 (SGLT2) inhibitors result in a 7.3% reduction in plasma volume without compensatory sympathetic nervous activation and improvement in renal function [24,25]. Future studies are required to examine whether PVS guided medication prevents premature deaths in the general population.

## Strengths and limitations

The strengths of the present study include its large sample size, prospective follow-up design, and nationwide data source. Therefore, our results are well generalized and highly reliable. However, there were some limitations as well. First, we assessed PVS at only one point in time. Second, we did not examine the incidence of non-fatal cardiovascular disease, cancer, and infectious disease and medical data. Thus, we could not determine the impact of PVS on the development of cardiovascular disease, cancer, and infectious disease. Third, we did not compare the calculated plasma volume but rather measured plasma volume in the general population. Finally, we do not have the data about physical activity, which may affect the future cardiovascular deaths.

## Conclusion

This is the first prospective report to reveal the impact of PVS on all-cause and cardiovascular mortality. Calculated PVS could be an additional risk factor for all-cause and cardiovascular mortality in the general population.

## Author Contributions

**Conceptualization:** Tsuneo Konta, Koichi Asahi, Ichiei Narita, Kunitoshi Iseki, Masahide Kondo.

**Data curation:** Yoichiro Otaki, Kazuhiko Tsuruya, Ichiei Narita, Kunitoshi Iseki.

**Formal analysis:** Tsuneo Konta, Ichiei Narita, Kunitoshi Iseki.

**Funding acquisition:** Masato Kasahara.

**Investigation:** Yoichiro Otaki, Shouichi Fujimoto, Masato Kasahara, Masahide Kondo.

**Methodology:** Tsuneo Konta, Koichi Asahi, Kunihiro Yamagata, Shouichi Fujimoto, Kazuhiko Tsuruya, Masato Kasahara, Toshiki Moriyama.

**Project administration:** Yoichiro Otaki, Koichi Asahi, Kunihiro Yamagata, Shouichi Fujimoto, Kazuhiko Tsuruya, Yugo Shibagaki, Toshiki Moriyama, Tsuyoshi Watanabe.

**Resources:** Koichi Asahi, Toshiki Moriyama.

**Supervision:** Masafumi Watanabe, Yugo Shibagaki, Masahide Kondo.

**Writing – original draft:** Yoichiro Otaki, Tetsu Watanabe.

**Writing – review & editing:** Yoichiro Otaki, Tetsu Watanabe, Masafumi Watanabe.

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
