## [Decision Letter · Decision Letter 0]

14 May 2020

PONE-D-20-07641

Impact of Calculated Plasma Volume Status on All-cause and Cardiovascular Mortality: 4-year Nationwide Community-Based Prospective Cohort Study

PLOS ONE

Dear Tetsu Watanabe

Thank you for submitting your manuscript to PLOS ONE. After careful consideration, we feel that it has merit but does not fully meet PLOS ONE’s publication criteria as it currently stands. Therefore, we invite you to submit a revised version of the manuscript that addresses the points raised during the review process.

We would appreciate receiving your revised manuscript by may-23. To enhance the reproducibility of your results, we recommend that if applicable you deposit your laboratory protocols in protocols.io, where a protocol can be assigned its own identifier (DOI) such that it can be cited independently in the future. For instructions see: http://journals.plos.org/plosone/s/submission-guidelines#loc-laboratory-protocols

We look forward to receiving your revised manuscript.

Kind regards,

Anderson Saranz Zago, PhD

Academic Editor

PLOS ONE

Additional Editor Comments (if provided):

According to the opinion of the reviewers, the manuscript brings an interesting subject, however, it needs to be reviewed on several topics. After all these changes, the authors can resubmit the manuscript for a new evaluation.

Sincerely

Anderson Saranz Zago, PhD.

Academic editor

2. In ethics statement in the manuscript and in the online submission form, please provide additional information about the patient records/samples used in your retrospective study. Specifically, please ensure that you have discussed whether all data/samples were fully anonymized before you accessed them and/or whether the IRB or ethics committee waived the requirement for informed consent. If patients provided informed written consent to have data/samples from their medical records used in research, please include this information.

3. We noticed you have some minor occurrence(s) of overlapping text with the following previous publication(s), which needs to be addressed:

https://doi.org/10.1253/circj.CJ-18-0721

In your revision ensure you cite all your sources (including your own works), and quote or rephrase any duplicated text outside the Methods section. Further consideration is dependent on these concerns being addressed.

4. Thank you for including your ethics statement:  "All procedures performed in studies involving human participants were undertaken in accordance with the ethical, institutional, and/or national research committee guidelines of the centers at which the studies were conducted (Yamagata University, 2008, no. 103) and in compliance with the 1964 Helsinki declaration and its later amendments or comparable ethical standards." 

a.) Please amend your current ethics statement to include the full name of the ethics committee/institutional review board(s) that approved your specific study.

b.) Please amend your current ethics statement to confirm that your named institutional review board or ethics committee specifically approved this study.

For additional information about PLOS ONE ethical requirements for human subjects research, please refer to " ext-link-type="uri" xlink:type="simple">http://journals.plos.org/plosone/s/submission-guidelines#loc-human-subjects-research."

"This work was supported by a Health and Labor Sciences Research Grant for Study on the design of the comprehensive health care system for chronic kidney disease (CKD) based on the individual risk assessment by Specific Health Check-up from the Ministry of Health, Labour and Welfare of Japan and a Grant-in-Aid for Research on Advanced Chronic Kidney Disease (REACH-J), Practical Research Project for Renal Disease from the Japan Agency for Medical Research and Development, AMED."

"The authors received no specific funding for this work."

7. Your ethics statement must appear in the Methods section of your manuscript. If your ethics statement is written in any section besides the Methods, please move it to the Methods section and delete it from any other section. Please also ensure that your ethics statement is included in your manuscript, as the ethics section of your online submission will not be published alongside your manuscript.

8. Please upload a new copy of Figure xxxx as the detail is not clear. Please follow the link for more information: https://blogs.plos.org/plos/2019/06/looking-good-tips-for-creating-your-plos-figures-graphics/

Reviewers' comments:

Reviewer's Responses to Questions

**Comments to the Author**

1. Is the manuscript technically sound, and do the data support the conclusions?

Reviewer #1: No

Reviewer #2: Yes

2. Has the statistical analysis been performed appropriately and rigorously? 

Reviewer #1: Yes

Reviewer #2: Yes

3. Have the authors made all data underlying the findings in their manuscript fully available?

Reviewer #1: Yes

Reviewer #2: Yes

4. Is the manuscript presented in an intelligible fashion and written in standard English?

Reviewer #1: Yes

Reviewer #2: Yes

5. Review Comments to the Author

Reviewer #1: The present study aimed to examine whether PVS is a novel risk factor for all-cause and cardiovascular deaths in the general population. It's an interesting study due to the variety of factors that could be related to cardiovascular mortality. However, the topic should be better explored.

The introduction section is really poor. It's not clear the relationship between plasma volume and mortality. What are the physiological mechanism that can explain this relationship ?

Why PVS is associated with cardiac events and mortality in patients with heart failure ?

Why is this discussion important and what are the hypothesis of the study?

Methods. What changes is expected during the aging process in the PVS ? This answer can explain if the age range (40 to 74 years old) is too high (or not).

In the “definition of PVS” section it is not clear where the values reported comes from (Men: a = 1530; b = 41; 3; c = 39 / women, a = 864; b = 47.9; c = 40). No reference is cited.

Is the authors used some statistical model to define the PVS range (7 and -13.3) ?

Is it possible a negative plasma volume ? It should be better explained.

How blood pressure were measured ?

Endpoint and follow-up section - All subjects were prospectively followed for a period of 914,292 person-years. (???????? Period should be in years/months/days). Participants were followed for 4 years. However the table one present the subject characteristics of the first year. How about the others ?

The results were presented in a difficult way understand.

In the page 7 the authors comment: higher PVS - older and low risk / low PVS - younger and high risk. However, in page 8 and 9, high PVS was significantly associated with all-cause, cardiovascular and non-cardiovascular deaths / page 10 - low PVS was significantly associated with all-cause and cardiovascular deaths, but not non-cardiovascular deaths.

Which is better ? High or low ?

The legend of figures are really poor. What does it means each line in the graphic ?

The interpretation of the graphics are really difficult. The legends do not provide such informations.

The first paragraph of discussion section shows a summary of the results. However the follow explanation it's not clean to a complete understanding of the relationship between PVS and cardiovascular risk.

Reviewer #2: The study now submitted is of good technical quality, its design was well prepared, presents extreme originality and offers great contributions to the application in the clinical area, including: early diagnosis, prognosis, emergency care and choices in making therapeutic decisions.

In addition, for a prospective epidemiological study, the Japanese territorial coverage was quite significant, Including more distant areas such as Okinawa, but no less important than the central areas.

However, on this point I have my first question: why did the researchers not include parts north of Honshu Island, in the representative Tohoku region, between Aomori and Fukoshima?

The second issue concerns possible differences in PVS between higher altitude regions. As is well known, the plasma volume as well as the hematocrit may vary significantly between inhabitants of regions close to sea level and mountainous regions. This could interfere with the PVS values, the mean PVS obtained, over or underestimating mean values, as well as interfere with the inference of the analogies.

In the case of a Japanese archipelago with a wide variation in altitude, the authors do not point out such aspects, homogeneity or not of the altitude of the regions covered by the study and also what measures were taken to minimize possible errors.

Another important question is, why were the mean PVS values of a random sample obtained from healthy people in the same age group and distributed in the regions of the study not obtained?

Finally, how do the authors justify the absence of the inclusion of the state of physical fitness or regular practice of physical activity or sedentary sedentary lifestyle among the variables analyzed and associated with PLWHA and its variations, since the regular practice of physical activity is recognized and widely used in the treatment of heart patients?

Could its effects on the patient's clinical status and consequently on PLWHA not interfere in the discussion of the results?

6. PLOS authors have the option to publish the peer review history of their article (what does this mean?). If published, this will include your full peer review and any attached files.

Reviewer #1: No

Reviewer #2: Yes: Cassiano Merussi Neiva

---

## [Author Response · Author response to Decision Letter 0]

22 May 2020

1. Is the manuscript technically sound, and do the data support the conclusions?

Reviewer #1: No

Reviewer #2: Yes

2. Has the statistical analysis been performed appropriately and rigorously?

Reviewer #1: Yes

Reviewer #2: Yes

3. Have the authors made all data underlying the findings in their manuscript fully available?

Reviewer #1: Yes

Reviewer #2: Yes

4. Is the manuscript presented in an intelligible fashion and written in standard English?

Reviewer #1: Yes

Reviewer #2: Yes

5. Review Comments to the Author

Reviewer #1: The present study aimed to examine whether PVS is a novel risk factor for all-cause and cardiovascular deaths in the general population. It's an interesting study due to the variety of factors that could be related to cardiovascular mortality. However, the topic should be better explored.

Q1. The introduction section is really poor. It's not clear the relationship between plasma volume and mortality. What are the physiological mechanism that can explain this relationship ? Why PVS is associated with cardiac events and mortality in patients with heart failure ? Why is this discussion important and what are the hypothesis of the study?

Answer to Q1. Thank you for your comments. We added the following sentences in Introduction section. 

‘Heart failure remains a major and increasing public health problem, with a high mortality rate [4]. It was reported that imbalanced volume homeostasis causes systemic congestion and peripheral and pulmonary edema in heart failure [5]. Plasma volume expansion underlies systemic congestion, which is a well-known, clinically, and prognostically relevant complication of heart failure [6]. (Page 3, line 3)

 ‘American College of Cardiology/American Herat Association guidelines have recommended volume status be assessed [16]. (Page 3, line 15)’

 ‘Thus, we hypothesized that PVS may serve as an early identification of high-risk subjects for all-cause and cardiovascular deaths in the general population. (Page 3, line 18)’

Q2. Methods. What changes is expected during the aging process in the PVS ? This answer can explain if the age range (40 to 74 years old) is too high (or not).

Answer to Q2. Thank you for your question. To the best of our knowledge, there was no report regarding the relationship between aging and PVS. There was a negative correlation between aging and actual PV and ideal PV in the present study. On the other hand, there was a weak, but significant positive correlation between aging and PVS. These findings suggested that plasma volume may shift toward volume expansion with aging. 

Q3. In the “definition of PVS” section it is not clear where the values reported comes from (Men: a = 1530; b = 41; 3; c = 39 / women, a = 864; b = 47.9; c = 40). No reference is cited.

Answer to Q3. Thank you for your suggestion. We added the citations.

Q4 Is the authors used some statistical model to define the PVS range (7 and -13.3) ?

Is it possible a negative plasma volume ? It should be better explained.

Answer to Q4. PVS is an index calculated by the following equation: (actual PV-ideal PV)/ideal PV, but not an estimated plasma volume itself. In TOPCAT study, PVS was reported to be less than 0 in 91% of heart failure patients (European Journal of Heart Failure. 2019; 21: 634-642).

Normal range of PVS has not been defined yet. In this study, we defined abnormally high and low PVS as PVS ≥ 7 and -13.3, respectively, based on the results for 80% of all subjects.

We added the following sentences in Method section

‘Since aPV is often under iPV, the value of PVS could become less than 0. In the Treatment of Preserved Cardiac Function Heart Failure with an Aldosterone Antagonist Trial (TOPCAT) study, PVS was reported to be less than 0 in 91% of heart failure patients [13]. High and low PVS are considered as plasma volume expansion and contraction, respectively. (Page 7, line 8)’

Q5. How blood pressure were measured ?

Answer to Q5. Thank you for your question. We added the following sentence.

 ‘Blood pressure was measured the following method [17]. Participants were seated with back supports. After resting for at least 5 minutes, blood pressure was measured 2 times without conversation. Blood pressure was determined by an average of 2 blood pressure readings. (Page 6, line 12)

Q6. Endpoint and follow-up section - All subjects were prospectively followed for a period of 914,292 person-years. (???????? Period should be in years/months/days). Participants were followed for 4 years. However the table one present the subject characteristics of the first year. How about the others ?

Answer to Q6. As you pointed out, we rewrote the follow up period as follows. Since the purpose of the present study was to examine whether baseline PVS can predict clinical outcome in the general population, we showed the baseline characteristics in study subjects in Table 1. 

‘All subjects were prospectively followed for a median follow up period of 4 years (interquartile range, 2.9-5.2 years; longest follow up, 7 years). (Page 8, line 1)

Q7. The results were presented in a difficult way understand.

In the page 7 the authors comment: higher PVS - older and low risk / low PVS - younger and high risk. However, in page 8 and 9, high PVS was significantly associated with all-cause, cardiovascular and non-cardiovascular deaths / page 10 - low PVS was significantly associated with all-cause and cardiovascular deaths, but not non-cardiovascular deaths.

Which is better ? High or low ?

Answer to Q7. High and low PVS mean imbalanced plasma volume in the general population. We considered the optimal PVS is associated with favor clinical outcome in the general population. 

Q8. The legend of figures are really poor. What does it means each line in the graphic ? The interpretation of the graphics are really difficult. The legends do not provide such informations.

Answer to Q8. According to your comments, we modified figure legend.

‘We collected data from 230,989 subjects (aged 40–74 years) who participated in the health check-ups of 2008–2011. Among them, 107 were excluded from this study due to lack of essential data. During the median follow-up period of four years, there were 586 cardiovascular deaths, 2,552 non-cardiovascular deaths, and 3,138 all-cause deaths. (Page 20, line 2)’

‘Solid line shows the hazard ratios for subject groups stratified by 1% increments of PVS. Dotted lines show the 95% confidence interval for subjects stratified by 1% increments of PVS. Orange bar shows the distribution of study subjects. The referent group was defined as the lowest risk group. *P0.05 v.s. referent group. (Page 20, line 7)

 ‘Green, red, and blue lines show the survival curves for high, normal, and low PVS groups, respectively. (Page 20, line 13)

Q9. The first paragraph of discussion section shows a summary of the results. However the follow explanation it's not clean to a complete understanding of the relationship between PVS and cardiovascular risk.

Answer to Q9. Thank you for your comments. We added the following sentences in Discussion section.

 ‘These findings indicated that imbalanced PVS may contribute to the development of cardiovascular disease. (Page 16, line 11)’

 ‘Potential explanation is that volume overload caused by plasma volume expansion may exacerbate cardiac function [6]. (Page 17, line 15)’ 

Reviewer #2: The study now submitted is of good technical quality, its design was well prepared, presents extreme originality and offers great contributions to the application in the clinical area, including: early diagnosis, prognosis, emergency care and choices in making therapeutic decisions.

Q1. In addition, for a prospective epidemiological study, the Japanese territorial coverage was quite significant, Including more distant areas such as Okinawa, but no less important than the central areas.

However, on this point I have my first question: why did the researchers not include parts north of Honshu Island, in the representative Tohoku region, between Aomori and Fukoshima?

Answer to Q1. Thank you for your question. Since we suffered from great earthquake in Tohoku region during study period, we could not obtain the follow-up data in this region unfortunately.

Q2. The second issue concerns possible differences in PVS between higher altitude regions. As is well known, the plasma volume as well as the hematocrit may vary significantly between inhabitants of regions close to sea level and mountainous regions. This could interfere with the PVS values, the mean PVS obtained, over or underestimating mean values, as well as interfere with the inference of the analogies.

In the case of a Japanese archipelago with a wide variation in altitude, the authors do not point out such aspects, homogeneity or not of the altitude of the regions covered by the study and also what measures were taken to minimize possible errors.

Answer to Q2. Thank you for your expertized question. We checked the residential area altitude. It ranged from sea level 31 m in Saitama to 644 m in Nagano. We found that there was a significant difference in PVS among region areas. Therefore, we added region area in multivariate analysis and calculation of C indices, NRI and IDI in order to adjust regional difference. PVS was still significantly associated with clinical outcomes and improved the prediction. We modified table 1, 2, and 3 and added the following sentences.

 ‘These prefectures were divided into four region areas; Hokkaido and Tohoku; Kanto and Koshinetsu; Kinki, Shikoku, and Chugoku; and Kyushu and Okinawa. (Page 6, line 5)

 ‘It was reported that high altitude affects volume status and its regulator hormone secretion [17], indicating the region difference in PVS. There was a significant difference in the prevalence of high and low PVS among region areas. (Page 11, line 20)’

Q3. Another important question is, why were the mean PVS values of a random sample obtained from healthy people in the same age group and distributed in the regions of the study not obtained?

Answer to Q3. Thank you for your comments. Since our study subjects were participants of health checkup, they were mainly apparently healthy subjects. Thus, we provide the data about PVS in healthy subjects.

Q4. Finally, how do the authors justify the absence of the inclusion of the state of physical fitness or regular practice of physical activity or sedentary sedentary lifestyle among the variables analyzed and associated with PLWHA and its variations, since the regular practice of physical activity is recognized and widely used in the treatment of heart patients? Could its effects on the patient's clinical status and consequently on PLWHA not interfere in the discussion of the results?

 Answer to Q4. Thank you for your suggestion. Unfortunately, we do not have the data about physical activity. Thus, we added the following sentence in Limitation section.

‘Finally, we do not have the data about physical activity, which may affect the future cardiovascular deaths. (Page 18, line 18)

6. PLOS authors have the option to publish the peer review history of their article (what does this mean?). If published, this will include your full peer review and any attached files.

Do you want your identity to be public for this peer review? For information about this choice, including consent withdrawal, please see our Privacy Policy.

Reviewer #1: No

Reviewer #2: Yes: Cassiano Merussi Neiva

---

## [Decision Letter · Decision Letter 1]

1 Jul 2020

PONE-D-20-07641R1

Impact of Calculated Plasma Volume Status on All-cause and Cardiovascular Mortality: 4-year Nationwide Community-Based Prospective Cohort Study

PLOS ONE

Dear Dr. Tetsu Watanabe

Thank you for submitting your manuscript to PLOS ONE. After careful consideration, we feel that it has merit but does not fully meet PLOS ONE’s publication criteria as it currently stands. Therefore, we invite you to submit a revised version of the manuscript that addresses the points raised during the review process.

As you can see both reviewer pointed that all comment were addressed. However, one of them made a comment about the legend of figure 1.

We look forward to receiving your revised manuscript.

Kind regards,

Anderson Saranz Zago, PhD

Academic Editor

PLOS ONE

Additional Editor Comments (if provided):

Both reviewer pointed that all comment were addressed. However, one of them made a comment about the legend of figure 1.

After all these changes, you can resubmit the manuscript for the final decision.

Reviewers' comments:

Reviewer's Responses to Questions

**Comments to the Author**

1. If the authors have adequately addressed your comments raised in a previous round of review and you feel that this manuscript is now acceptable for publication, you may indicate that here to bypass the “Comments to the Author” section, enter your conflict of interest statement in the “Confidential to Editor” section, and submit your "Accept" recommendation.

Reviewer #1: All comments have been addressed

Reviewer #2: All comments have been addressed

2. Is the manuscript technically sound, and do the data support the conclusions?

Reviewer #1: Yes

Reviewer #2: Yes

3. Has the statistical analysis been performed appropriately and rigorously? 

Reviewer #1: Yes

Reviewer #2: Yes

4. Have the authors made all data underlying the findings in their manuscript fully available?

Reviewer #1: Yes

Reviewer #2: Yes

5. Is the manuscript presented in an intelligible fashion and written in standard English?

Reviewer #1: Yes

Reviewer #2: Yes

6. Review Comments to the Author

Reviewer #1: The manuscript has substantially improved after review. It can be observed that all comments have been addressed. However, I would make a last suggestion, given the changes made in the legend of the figures.

In the legend of figure 1, the authors included the following sentence: “We collected data from 230,989 subjects (aged 40–74 years) who participated in the health check-ups of 2008–2011. Among them, 107 were excluded from this study due to lack of essential data. During the median follow-up period of four years, there were 586 cardiovascular deaths, 2,552 non-cardiovascular deaths, and 3,138 all-cause deaths” This sentence should not be included in the legend since it represent a description and it is already included in the study population section.

Reviewer #2: I have no further questions and consider that the authors have answered satisfactorily the questions asked previously, as well as the suggested corrections.

7. PLOS authors have the option to publish the peer review history of their article (what does this mean?). If published, this will include your full peer review and any attached files.

Reviewer #1: **Yes: **Anderson Saranz Zago

Reviewer #2: **Yes: **Cassiano Merussi Neiva

---

## [Author Response · Author response to Decision Letter 1]

2 Jul 2020

Comments to the Author

1. If the authors have adequately addressed your comments raised in a previous round of review and you feel that this manuscript is now acceptable for publication, you may indicate that here to bypass the “Comments to the Author” section, enter your conflict of interest statement in the “Confidential to Editor” section, and submit your "Accept" recommendation.

Reviewer #1: All comments have been addressed

Reviewer #2: All comments have been addressed

2. Is the manuscript technically sound, and do the data support the conclusions?

Reviewer #1: Yes

Reviewer #2: Yes

3. Has the statistical analysis been performed appropriately and rigorously?

Reviewer #1: Yes

Reviewer #2: Yes

4. Have the authors made all data underlying the findings in their manuscript fully available?

Reviewer #1: Yes

Reviewer #2: Yes

5. Is the manuscript presented in an intelligible fashion and written in standard English?

Reviewer #1: Yes

Reviewer #2: Yes

6. Review Comments to the Author

Reviewer #1: The manuscript has substantially improved after review. It can be observed that all comments have been addressed. However, I would make a last suggestion, given the changes made in the legend of the figures.

In the legend of figure 1, the authors included the following sentence: “We collected data from 230,989 subjects (aged 40–74 years) who participated in the health check-ups of 2008–2011. Among them, 107 were excluded from this study due to lack of essential data. During the median follow-up period of four years, there were 586 cardiovascular deaths, 2,552 non-cardiovascular deaths, and 3,138 all-cause deaths” This sentence should not be included in the legend since it represent a description and it is already included in the study population section.

Answer to Q1. According to your suggestion, we deleted it. We express our sincere thanks to your effort.

Reviewer #2: I have no further questions and consider that the authors have answered satisfactorily the questions asked previously, as well as the suggested corrections.

Answer to Reviewer #2. We express our sincere thanks to your effort.________________________________________

7. PLOS authors have the option to publish the peer review history of their article (what does this mean?). If published, this will include your full peer review and any attached files.

Do you want your identity to be public for this peer review? For information about this choice, including consent withdrawal, please see our Privacy Policy.

Reviewer #1: Yes: Anderson Saranz Zago

Reviewer #2: Yes: Cassiano Merussi Neiva

---

## [Decision Letter · Decision Letter 2]

30 Jul 2020

Impact of Calculated Plasma Volume Status on All-cause and Cardiovascular Mortality: 4-year Nationwide Community-Based Prospective Cohort Study

PONE-D-20-07641R2

Dear Dr. Watanabe,

We’re pleased to inform you that your manuscript has been judged scientifically suitable for publication and will be formally accepted for publication once it meets all outstanding technical requirements.

Kind regards,

Anderson Saranz Zago, PhD

Academic Editor

PLOS ONE

Additional Editor Comments (optional):

I am pleased to inform you that your manuscript has been deemed suitable for publication in PLOS ONE. Congratulations!

Sincerely

Reviewers' comments:

Reviewer's Responses to Questions

**Comments to the Author**

1. If the authors have adequately addressed your comments raised in a previous round of review and you feel that this manuscript is now acceptable for publication, you may indicate that here to bypass the “Comments to the Author” section, enter your conflict of interest statement in the “Confidential to Editor” section, and submit your "Accept" recommendation.

Reviewer #1: All comments have been addressed

2. Is the manuscript technically sound, and do the data support the conclusions?

Reviewer #1: Yes

3. Has the statistical analysis been performed appropriately and rigorously? 

Reviewer #1: Yes

4. Have the authors made all data underlying the findings in their manuscript fully available?

Reviewer #1: Yes

5. Is the manuscript presented in an intelligible fashion and written in standard English?

Reviewer #1: Yes

6. Review Comments to the Author

Reviewer #1: (No Response)

7. PLOS authors have the option to publish the peer review history of their article (what does this mean?). If published, this will include your full peer review and any attached files.

Reviewer #1: **Yes: **Anderson Saranz Zagp

---

## [Editor Report · Acceptance letter]

4 Aug 2020

PONE-D-20-07641R2 

Impact of Calculated Plasma Volume Status on All-cause and Cardiovascular Mortality: 4-year Nationwide Community-Based Prospective Cohort Study 

Dear Dr. Watanabe:

I'm pleased to inform you that your manuscript has been deemed suitable for publication in PLOS ONE. Congratulations! Your manuscript is now with our production department. 

Kind regards, 

on behalf of

Dr. Anderson Saranz Zago 

Academic Editor

PLOS ONE